# Population-based cohort across stroke life course in India-The NIMHANS-NH-SKAN stroke project: A study protocol

Pradeep S. Banandur[1]*, Gautham Melur Sukumar[1], Banavaram Anniappan Arvind[1], Srijithesh P. R.[2], Binu V. S.[3], Santosh Loganathan[4], Nandakumar Dalavaikodihalli Nanjaiah[5], Thimappa Hegde[6], Komal Prasad[7], Lavanya Garady[8], Akashanand Akashanand[1], Palak Poddar[1], Meenakshi Jayabalan[1], Isha Madan[1], Upashana Medhi[9], Arpitha Arun[9]

1 Dept. of Epidemiology, NIMHANS, Bengaluru, Karnataka, India, 2 Department of Neurology, NIMHANS, Bengaluru, Karnataka, India, 3 Department of Biostatistics, NIMHANS, Bengaluru, Karnataka, India, 4 Department of Psychiatry, NIMHANS, Bengaluru, Karnataka, India, 5 Department of Neurochemistry, NIMHANS, Bengaluru, Karnataka, India, 6 Mazumdar Shaw Medical Centre, Narayana institute of Neurosciences, Bengaluru, Karnataka, India, 7 Narayana Institute of Neurosciences, Bengaluru, Karnataka, India, 8 SKAN Research Trust, Bengaluru, Karnataka, India, 9 Skan Project, NIMHANS, Bengaluru, Karnataka, India

* doctorpradeepbs@gmail.com

**Data Availability Statement:** No datasets were generated or analyzed during the current study. All relevant data from this study will be made available upon study completion. All data collected will be uploaded into a password protected (3-layers)

## Abstract

### Background

Stroke is a leading cause of death and disability worldwide. In India, it is the fourth leading cause of death and fifth leading cause of disability, posing a major public health concern. National surveys reveal an increasing trend in stroke risk factors such as tobacco use, physical activity, alcohol use, hypertension, and dyslipidemia. However, knowledge regarding the combined effect of these risk factors and their various combinations is limited. Understanding the individual, combined, and synergistic effects of known risk factors, along with new risk factors, is essential to address gaps in stroke epidemiology. This study aims to examine the effect of various risk factors of acute stroke and their association with stroke occurrence and its outcomes (survival, disability and quality of life).

### Methods

This retrospective-prospective cohort will be conducted in one taluka of Kolara district and two urban wards of Bengaluru with a total population of ~400,000. All stroke-free individuals above 30 years of age ~200,000 individuals in the selected sites will be participants of stroke-free period and all first ever stroke patients in the community will be part of stroke and post-stroke period respectively. The study subjects will be recruited through a complete house-to-house survey at baseline and undergo annual follow-ups during the stroke-free period, with specific assessments at defined time points during the stroke and post-stroke period for a period of one year. Efforts are implemented to minimize loss to follow-up, including community engagement, a helpline number, and hospital-based surveillance.

server with access only to specified authorized study team members. Every access and activity within the server will be logged as per pertinent policies, rules and guidelines.

**Funding:** The study is funded by non-commercial organization SKAN research trust URLs to funder's websites: https://skanrt.in/ While our research project is funded, I wish to emphasize that our research endeavours are entirely non-commercial in nature and are dedicated to advancing academic knowledge within our respective field. Authors- Dr. Thimappa Hegde, Dr. Komal Prasad, and Dr. Lavanya Garady contributed to the conceptualization of the study and reviewed and finalized the manuscript.

**Competing interests:** The authors have declared that no competing interests exist.

## Discussion

This large population-based cohort study addressing stroke epidemiology in the country, is one -of-its-kind, attempting to fill certain critical gaps in the natural history, management, and outcomes of stroke in India. This research has the potential to provide important insights into the effect of novel risk factors of stroke and various combinations of risk factors of stroke. Furthermore, the development of a stroke risk predictability calculator will add value to the existing Indian National Programme for Prevention & Control of Non-Communicable Diseases (NP-NCD) and offers a model for similar countries once developed.

## Introduction

Increasing life expectancy in most countries has resulted in increased proportion of elderly and associated burden of chronic Non-Communicable Diseases (NCDs). NCDs account for about two-thirds of deaths both globally (68%) and in India (60%) [1] leading to significant long term health, economic and social costs. The proven cost-effectiveness of prevention interventions, make NCD prevention and control, a priority in the 21st century [2].

Stroke is the second leading cause of death and the leading cause of disability [2, 3]. Globally, between 1990 and 2019, there is a 70% increase in stroke incidence, a 43% rise in stroke-related deaths, a 102% increase in stroke prevalence, and 143% surge in Disability-Adjusted Life Years (DALYs) [4]. India, is currently undergoing an epidemiological transition from communicable diseases to NCDs [5]. Stroke is the fourth leading cause of death, fifth leading cause of disability [6], and a major public health concern in India. In addition to being fatal, stroke is associated with short and lifetime disability among survivors, affecting their quality of life and productivity [7]. A high proportion of stroke survivors suffer from permanent impairments resulting in deficient self-care, requiring long-term support from care-givers, adding to individual health and social costs [8].

Epidemiology and care of stroke widely varies across settings in India. Stroke accounted for 7.73% fatalities and 4.26% of DALYs in Karnataka, a state in southern India [9]. An estimated 500,000 stroke cases are prevalent at any given point of time in Karnataka, placing immense burden on the health system [9]. The situation is compounded by lack of regular, valid and reliable stroke related epidemiological and treatment data for evidence-driven programming [10]. Incidence of stroke is largely dependent on incidence of stroke risk factors, which are dynamic and defined based on understanding of epidemiology, from other countries.

Ongoing national surveys and research studies indicate an increasing trend in prevalence of known stroke risk factors like tobacco use, physical inactivity, alcohol use, hypertension and dyslipidemia. However, current knowledge lacks clarity on the combined effect of multiple risk factors and their various combinations. Understanding the individual, combined and synergistic effect of these known stroke risk factors coupled with information on neo-risk factors for stroke, would help to bridge gaps in stroke epidemiology. This enables in-depth understanding of stroke and its risk factors and facilitates evidence-based health systems and services for risk reduction, case management and rehabilitation of stroke.

The NIMHANS-NH-SKAN stroke project, a large population-based cohort study, aims to examine all issues related to stroke covering risk factors, occurrence, treatment and outcomes. This project also intends to develop an India-specific stroke risk predictability calculator for the studied risk factors that shall predict an individual's risk of stroke over specific time

periods. The NIMHANS-NH-SKAN stroke project is implemented by the Department of Epidemiology, Centre for Public Health, NIMHANS, Bengaluru

We describe the methodology of NIMHANS-NH-SKAN Stroke project which aims to:

1. estimate the incidence and incidence rates of first-ever stroke among risk factors (individually and combined) in Karnataka.

2. estimate the strength of association between select risk factors and acute first-ever stroke in Karnataka.

3. estimate short- and long-term survival, disability and quality of life of first-ever stroke patients in Karnataka.

4. identify and estimate strength of association of factors associated with short- and long-term survival, disability and quality of life of first-ever stroke patients in Karnataka.

5. develop a specific risk predictability calculator for development of stroke over specific time periods.

## Methodology

This population-based cohort study assesses the effect of various risk factors on stroke occurrence and its outcomes namely survival, disability and quality of life. For the ease of understanding methodology is explained in two parts (Fig 1):

1. Stroke-free period–period from initial recruitment of a study subject until development of stroke or end of follow-up period.

2. Stroke and post-stroke period–period after when a study subject develops stroke until one-year post-stroke or death whichever is earlier.

### Study design and study settings

**Stroke-free period.** This is a retrospective-prospective cohort study (mixed/ ambispective cohort study) conducted in two urban wards of Bengaluru city and one taluka (sub-district) of Kolara district in Karnataka, a southern Indian state. Specific wards and sub-districts were selected for operational convenience. Our strong presence and established rapport within these communities will enable easy acceptance of the survey and effective community mobilization. This approach is expected to result in higher response rates and minimal lost to follow-up. Naturally existing risk groups constitute exposures and stroke as outcome.

**Stroke and post-stroke period.** This is a prospective cohort study; we will enroll all first-time stroke cases occurring in the study area. Stroke and type of stroke constitute exposure. Survival, disability and quality of life constitutes outcome.

### Identification and recruitment of study subjects

**Stroke free period.** All stroke-free permanent residents currently residing in the study area aged 30 years and above, and willing to participate will be included. Individuals with previous history of stroke and chronic debilitated/ bed ridden individuals or those who are otherwise unable to participate in the study will be excluded from the study. A baseline house-to-house enumeration of all eligible study subjects will be conducted in the study area. Eligible

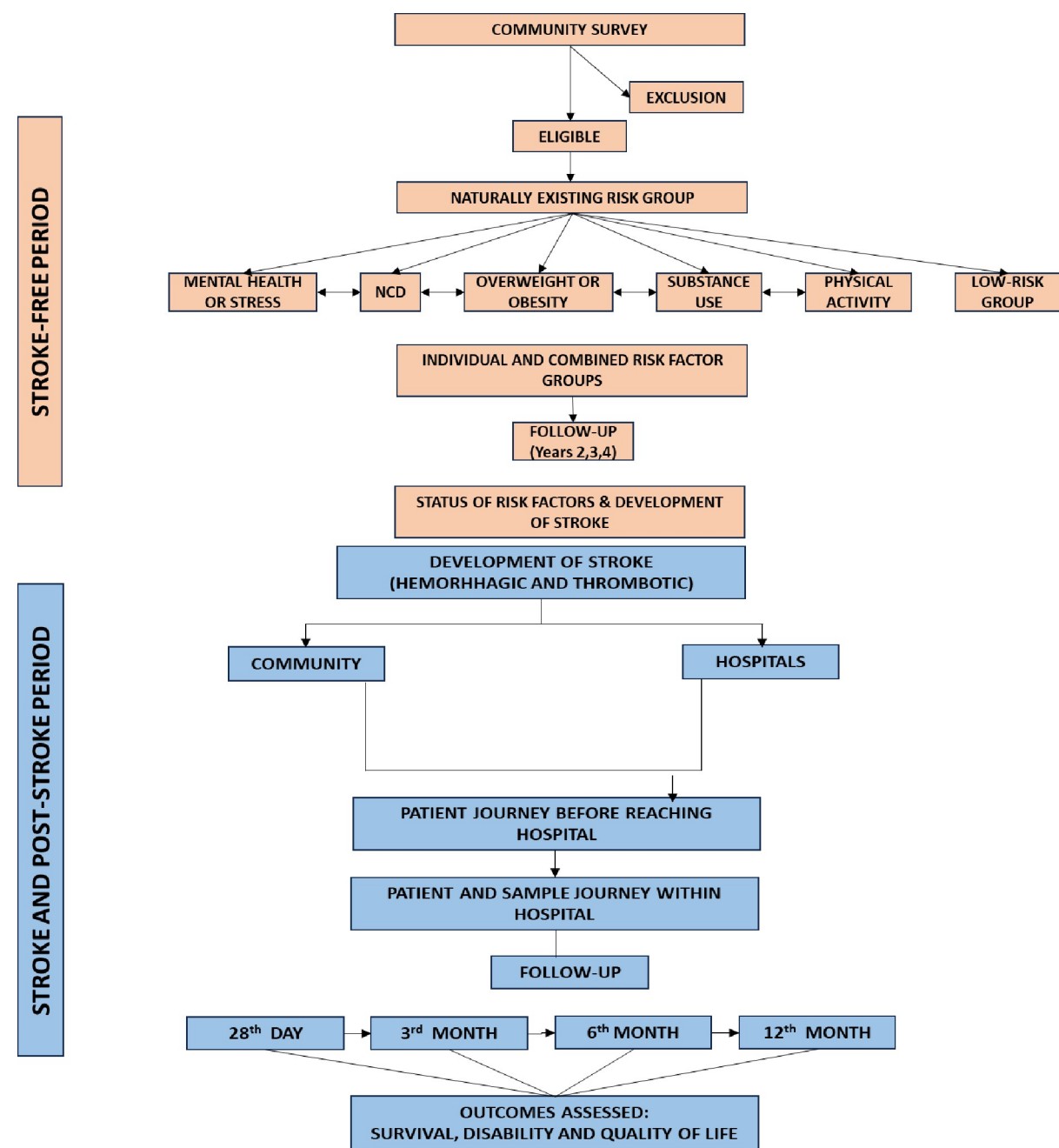

**Fig 1. Overview of the NIMHANS-NH-SKAN stroke project.**

respondents who enter the study area during the first and second year of follow-up will also be included in the study. An IEC/Community engagement plan to actively engage and mobilize the community throughout the project period is being developed. The strategy is to create awareness and disseminate information regarding the project to the community, their participation and for information to the study team about the cases occurring within the study areas. In addition, this community engagement is likely to minimize non-response as well as dissolve

any issues that might arise due to misconceptions regarding the study that influence participation.

**Stroke and post-stroke period.** All first ever stroke cases reported within the study area shall be recruited. Primarily, recruitment will be within the hospitals where the study subjects are admitted. All these cases will be identified through community informants (family members, health workers/ASHAs/Anganwadi workers/community leaders/hospital staff/ anybody who are likely to know the occurrence of stroke) within the study area. All field workers and community will be sensitized and trained to report the occurrence of stroke and will be compensated for factual reporting. As part of community engagement, an exclusive helpline will be established to enable case identification. Hospital based surveillance will also be setup to identify cases from the study area admitted in the hospitals.

## Setting and healthcare available

India's healthcare system operates on a three-tier hierarchical model. At the primary level, basic health services are offered through Primary Health Centres and Sub-Centres (Health and Wellness Centres) located in villages or towns. The secondary level includes district hospitals that provide additional diagnostic and surgical services along with basic healthcare. Tertiary care is offered by medical colleges and hospitals, which deliver advanced medical treatments and surgeries. Stroke cases typically receive care at either the secondary or tertiary levels.

Based on previous records, study subjects from the study area mostly visit a network of one neuro specialty hospital and one medical college hospital at tertiary level and one district hospital, one medical college hospital and two government hospitals at the secondary level.

Alternatively, if the study subjects are admitted in hospitals which are not within reasonable distance, recruitment shall be done after they come back from the hospitals within their households.

## Follow-up

**Stroke-free period.** After the baseline assessment, all survey respondents will be followed up once annually at three time points (year 2, year 3 and year 4) (Fig 2). For those respondents who move out of the study area, attempts will be made to collect data from them, if they are available within a reasonable travel distance (e.g.: within Kolara and close by urban wards within Bengaluru). Efforts will also be made through telephonic calls either to contact them personally or ascertain their stroke status, to the extent possible.

The study end-point for subjects in the stroke-free period will be when the subject (Fig 2)

1. completes the follow-up period of four years without developing stroke

2. develops stroke during the four-year follow-up period

3. dies/ migrates from the study area/ is loss to follow-up

4. stops participation in the study for any other reason

**Stroke and post-stroke period.** Any respondent who develops stroke during the study period will be assessed within the hospital and followed up at four time-points (28th day, 3rd month, 6th month and 12th month) post-stroke (Fig 2). For those respondents who move out of the study area, efforts similar to stroke free period shall be made to minimize loss to follow up.

The study end-points for post-stroke period subjects will be when the subject

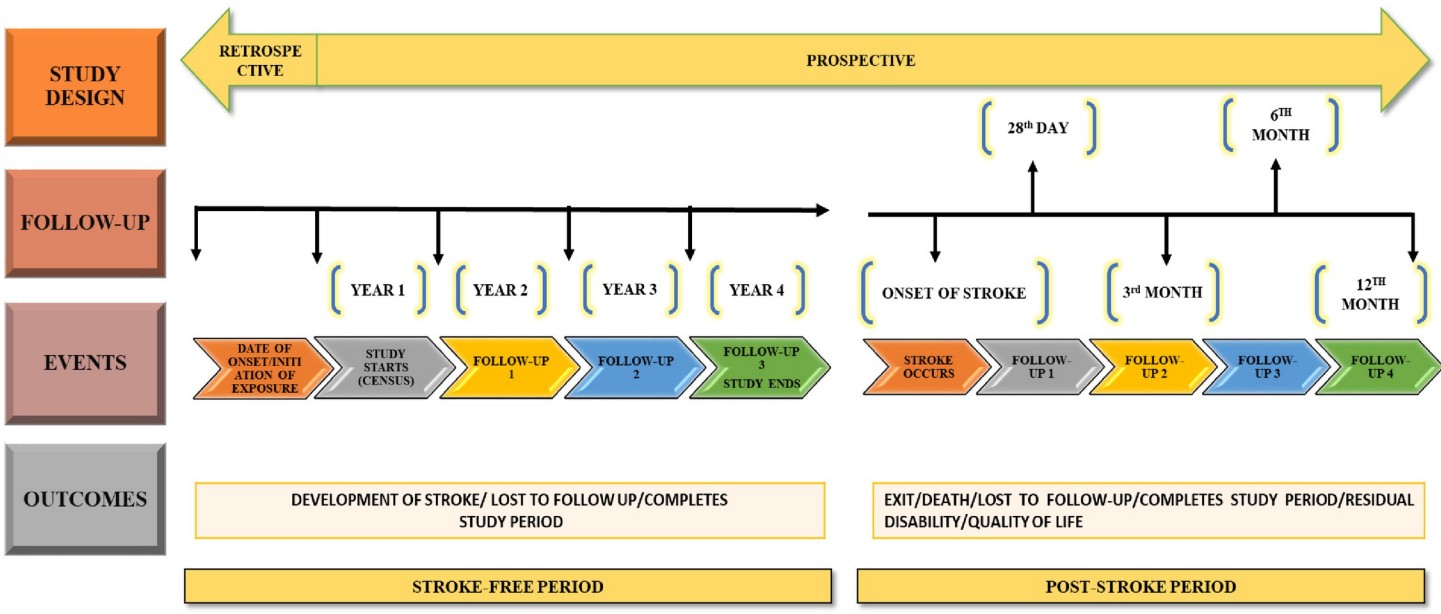

**Fig 2. Schematic representation of NIMHANS-NH-SKAN stroke project.**

1. completes the follow-up period of 1 year

2. dies/migrates/ lost to follow-up

3. stops participation in the study for any other reason

## Ascertainment of exposure and outcome

**Stroke-free period.**   We plan to study multiple exposures in this study. Naturally existing stroke risk groups namely weight, mental health/ stress (includes depression, anxiety, stress, non-suicidal self-injury, suicidal behavior and workplace stress), non-communicable diseases (diabetes mellitus, hypertension, dyslipidemia and coronary vascular diseases) substance use including tobacco (smoking/smokeless), alcohol and drugs (injecting/sniffing/oral), nutrition, physical inactivity and certain neo-risk factors namely gambling, COVID-19 infection and vaccination, sleep (insomnia and obstructive sleep apnea), exposure to media, risk of cell phone addiction and Restless leg syndrome constitute exposures. The primary outcome assessed is stroke (including its sub-types).

**Stroke and post-stroke-period.**   We plan to compare/assess the difference in the survival (both long term and short term), disability and quality of life among stroke patients. Type of stroke will be considered *apriori* exposure along with different exposures assessed for the stroke free period as confounders.

The definitions/ assessments proposed to ascertain different exposures and outcome is shown in Table 1.

## Study instruments

All eligible participants will be interviewed using semi structured partially open-ended schedules specifically developed for data collection and posted on low-literacy-user-friendly digital platform. These digital schedules are separate for stroke-free and stroke and post-stroke

**Table 1. Instruments used for data collection NIMHANS-NH-SKAN stroke project.**

| Name of the schedule | Contents/scale used | Study subjects | Time point of data collection | Period |
|---|---|---|---|---|
| **Census schedule** | Household information including number of residents, assets and material holding along with socio-demographic and economic characteristics of all individuals of the household. | Head of the household or any responsible resident | Once a year | Stroke-free |
| **Baseline schedule** | Detailed information on risk factors, their date of onset/diagnosis, duration, dose or amount of risk factor as appropriate (example; average number of cigarettes smoked/day, alcohol consumed/day etc.), current status and dependence information (as appropriate). | All eligible respondents in the study area | First Year only | Stroke-free |
| **Follow-up schedule** | Screening questions related to each risk factor, current status (during the interim period) and dependence information (as appropriate). | All eligible respondents in the study area | Every follow-up | Stroke-free and post and post-Stroke |
| **Quality of life schedule** | WHO quality of life (QOL) detailed schedule to assess all domains of quality of life (Physical, social, psychological, economic and spiritual domains of quality of life. | All eligible respondents in the study area | Every follow-up+ Baseline | Stroke-free |
| | SS-QOL is a detailed schedule assessing 12 domains of energy, family roles, language, mobility, mood, personality, self-care, social roles, thinking, upper extremity (UE) function, vision, and work/productivity | All individuals who develop stroke within the study area | Every follow-up | Stroke and post-stroke |
| **Outcome schedule** | Information related to development of stroke, disability scale (standardized, validated disability assessment scale) and survival including information related to medication adherence, post hospitalization care | All individuals who develop stroke within the study area | Every follow-up | Stroke and post-stroke |
| **Management and pathways of care schedule** | Includes case record forms with information related to history, clinical information, laboratory investigations conducted and management procedures performed and drugs prescribed. Pathways to care schedule consists of information on the symptoms reported, time taken to reach the healthcare facility, type of health services sought, number of healthcare facilities visited for treatment and health expenditure on the same both within the hospital and before reaching the hospital. All these recorded from hospital records/documentation. | All individuals who develop stroke within the study area | As and when individuals within the study area develop stroke | Stroke and post-stroke |

period. Each of these schedules will have multiple forms with multiple sections. The contents and scales utilized are standardized validated scales approved by the expert advisory group specifically constituted for the project (Table 1).

## Data collection

Data collection shall be done using a combination of face-to-face interview utilizing a specifically developed semi-structured partially open-ended interview schedule, review of clinical records and biological investigations. Initially, a complete household census will be done in the study area to identify eligible study respondents. The interview schedule consists of 30 sections (S1 Table). Data on socio demographic, clinical history, family medical history, in addition to participant reported exposures, information on disability and quality of life of the participant/patient will be obtained using data collected during the stroke-free period. Data collection will be performed by trained individuals with previous experience in population and/ hospital based medical research data collection, community and social work. The trained data collectors will also measure height, hip abdominal circumference and blood pressure (measured after 15 minutes of resting in a comfortable sitting posture). Biological samples will be collected by trained laboratory investigators, stored and transported to the laboratories immediately. Samples from district will be sent to the medical college hospital or district hospital. Sample from Bengaluru will be sent to the Neuro-Specialty care hospital. All data will be

collected using a low-literacy user-friendly digital data collection instrument specifically developed for the project. The field supervisors/coordinators along with program coordinators will perform 5% repeat data collection for assessment of quality of data collected. Any deviation/errors in data collection more than 10%, calls for repeat survey within the respective data collection area.

During data collection for stroke and post-stroke period, care-givers of subjects will be interviewed if the participants are unable to provide information. The participants/caregivers will be assisted by the data collectors in responding to the questionnaire. In case of death of the participant during the study period, the date of death along with other appropriate/relevant information (through medical records and caregiver information) will be obtained. For those respondents who move out of the study area, attempts will be made to collect data from them either face-to-face, if they are available within a reasonable travel distance (e.g.: within Kolara and close by urban wards within Bengaluru) or through telephonic interview.

### Training of data collection

The training for the data collectors and laboratory investigators will be participatory, employing different methods including classroom sessions, training in the hospital (observation and demonstration of interviews), community (both supervised and independent) and hands-on training (for digital data collection and biological sample collection) by specialized personnel for each domain. It is planned to conduct induction and refresher trainings at the beginning of each data collection cycle (once a year) for the study team ensuring standardized data collection. To ensure quality apart from rigorous training, weekly and fortnightly review and problem-solving meetings will be held both locally and with the NIMHANS-NH-SKAN team online.

### Monitoring

An independent monitoring team (IMT) is constituted to oversee project implementation activities. Members of the team include mid-level specialists mainly from the field of Community medicine/ Epidemiology/Public Health, Neurology and Social work with relevant work experience. The IMT will maintain a constant and ongoing supervision to ensure that the project activities are carried out as per protocol. The IMT will conduct monitoring visits every quarter and submit a report to the study team emphasizing the progress and identifying critical bottle-necks/challenges in the implementation of the study.

### Expert advisory group

The Expert Advisory Group (EAG), composed of specialists in Neurology, Neurosurgery, Epidemiology, Biostatistics, and Public Health with extensive research experience in Public Health and Stroke Epidemiology, is established to provide strategic oversight and technical support for the study. This group will play a crucial role in directing the study, monitoring its progress, ensuring its quality, and reviewing timelines. The group will provide regular and timely advice to study investigators on the study's progress and played an integral role in reviewing the master protocol and suggesting modifications. The group convenes biannually (every 6 months) to assess the study's progress and provide necessary input.

### Data management

All data collected will be uploaded into a password protected (3-layers) server with access only to specified authorized study team members. Every access and activity within the server will be

logged as per pertinent policies, rules and guidelines. The core team within NIMHANS shall check for completeness, quality and accuracy of data collected utilizing specifically developed quality check formats. Any discrepancy/ clarifications shall be sought with the field team and rectified/ modified within the server on a weekly basis. Only specific core team members authorized to access within NIMHANS shall have the access and rights to modify data as per field team recommendations/justification. All the modifications shall be time logged. All forms, data collected and stored will be linked through unique ID of the study participants including laboratory and follow-up data.

## Biochemical sample collection and measurements

Laboratory confirmation for estimation of glucose and lipid profile will be done using venous blood sample collected from study participant. Approximately 5–7 ml of venous blood will be collected from the cubital vein using a needle and syringe. Blood sample will be drawn by qualified/trained personnel under aseptic precautions using disposable needles. Adequate pressure will be applied for sufficient duration of time to prevent bleeding from the site of blood collection. Immediately, the collected blood will be transferred to plasma separator vacutainer tubes. Plasma will be separated after centrifugation. Serum glucose and lipid profile will be estimated using hexokinase method and colorimetric assay respectively.

## Sample size and power calculations

**Stroke-free period.**   The study sites constitute two urban wards in Bengaluru and Mulbagal taluka in Kolara district (with 20:80 –urban: rural distribution of population). Each urban ward is expected to have a minimum of 80,000 people. Approximate population within the taluka is 250,000. The proportion of population above 30 years is approximately 58.9% [11]. Thus, about 241,490 subjects (410,000*0.589) are expected to be available for this project. Assuming 20% attrition, we expect ~193,192 subjects to participate in the study.

Assuming 150 new stroke cases per 100,000 population per year and follow-up duration of 3 years at 5% level of significance, sample sizes are calculated for different risk group (Table 2). Since the expected prevalence of obesity is the least among all the risk groups (6.2%), sample size estimation based on obesity as exposure is expected to cover the required sample size for other groups also. With the expected population of ~240,000 subjects, the study results would have a power of 90% to achieve a Hazard Ratio (HR) of 1.4. Even with an attrition rate of 20% (~193,000 subjects) we would be able to achieve the proposed objective with >80% power.

**Stroke and post-stroke period.**   With the minimum estimated number of 193,000 respondents available (after accounting for 20% attrition) for stroke-free cohort and an expected incidence of 150 per 100,000 population per year, we expect around 290 cases of stroke occurring in the study areas every year. Thus, a total of 870 cases are expected to occur in the study area during the study period. Efforts shall be made to recruit all these cases into the stroke cohort study.

## Statistical analysis

**Stroke-free period.**   Study subjects will be assigned into different risk factor groups. Multiple individual and combination of naturally existing exposure groups within the community shall be formed. The specific risk groups (comparison groups) of single, dual, triple or multiple risk factors for stroke are formed to estimate and compare the incidence of stroke among multiple exposure groups. Formation, classification, and follow-up of risk factor groups would be made in the database. Neither the study subjects nor the data collectors shall be informed about the grouping of their specific risk groups. However, information regarding the health status shall be provided to the respondent to enable them to seek appropriate care.

**Table 2. Sample size calculation for different Hazard ratios and power among apriori risk factor groups.**

| Risk groups | Hazard ratio, power | Expected population prevalence | | |
|---|---|---|---|---|
| | | **Exposed** | **Unexposed** | **Total** |
| **Nutrition** [12]: 6% for obesity | 1.4, 80% | 9700 | 151960 | 161660 |
| | 1.4, 90% | 13363 | 209347 | 222709 |
| | 1.5, 80% | 6370 | 99797 | 106167 |
| | 1.5, 90% | 8828 | 13830 | 147128 |
| **Mental health/Stress** [13]: 11% for current mental morbidity | 1.4, 80% | 10328 | 83560 | 93888 |
| | 1.4, 90% | 14180 | 114733 | 128913 |
| | 1.5, 80% | 6796 | 54987 | 61784 |
| | 1.5, 90% | 9380 | 75893 | 85273 |
| **NCD** [14]: 29% for raised blood pressure | 1.4, 80% | 13307 | 32580 | 45888 |
| | 1.4, 90% | 18058 | 44211 | 62269 |
| | 1.5, 80% | 8814 | 21579 | 30393 |
| | 1.5, 90% | 11993 | 29361 | 41354 |
| **NCD** [14]: 10% for raised blood sugar | 1.4, 80% | 10197 | 91769 | 101965 |
| | 1.4, 90% | 14010 | 126088 | 140098 |
| | 1.5, 80% | 9708 | 60365 | 67073 |
| | 1.5, 90% | 9265 | 83384 | 92648 |
| **NCD** [14]: 13% for existing CVD | 1.4, 80% | 10599 | 70930 | 81528 |
| | 1.4, 90% | 14533 | 97262 | 111795 |
| | 1.5, 80% | 6980 | 46712 | 53692 |
| | 1.5, 90% | 9618 | 64368 | 73986 |
| **Substance use** [14]: 33% for current tobacco use | 1.4, 80% | 14184 | 28978 | 42982 |
| | 1.4, 90% | 19198 | 38978 | 58176 |
| | 1.5, 80% | 9407 | 19019 | 28506 |
| | 1.5, 90% | 12759 | 25905 | 38665 |
| **Substance use** [14]: 16% for current alcohol use | 1.4, 80% | 11029 | 57903 | 68932 |
| | 1.4, 90% | 15094 | 79241 | 94335 |
| | 1.5, 80% | 7272 | 38176 | 45448 |
| | 1.5, 90% | 9996 | 52480 | 62476 |

*Expected population prevalence of risk factors are rounded off to the nearest round number. Sample size calculations are based on the formula given by [15].

Incidence proportion and Incidence rates for stroke and different types of stroke will be calculated and presented with corresponding 95% confidence intervals. Actual life table methods will be used for estimation on Incidence rates. Person-months of follow-up and person-months of exposures (for each risk factor) will be estimated. Overall incidence rates for person-months of follow-up will be estimated. Incidence rate for specific exposure groups will use person-months/ person-years of exposure to risk factor as the denominator. Measures of risk for stroke and type of stroke, for each risk factor, will be determined using appropriate regression methods. Attributable and population attributable risk (PAR) and the corresponding 95% CIs will be obtained. For those study subjects who prematurely exit (dies/ migrates from the study area/ lost to follow-up or stops participation for any other reason) from the study, the stroke-free duration of such participants will be utilized for analysis.

The risk factors namely weight, mental health/stress, NCDs, substance use, nutrition and physical inactivity are considered *apriori* exposures and stroke as outcome. A conceptual framework based on multifactorial web of causation developed by Mc Mohan and Pugh [16] consisting of *apriori* exposures and other potential confounders influencing the association

between them and the outcome will be developed through desk review and expert consultation. This shall form the basis for statistical analysis for the study. The difference in stroke risk for the different risk groups will be assessed using the Cox proportional hazard models with age of onset of risk factor (e.g., start of smoking/tobacco use/ alcohol etc.) as the time scale. Stroke as well as its sub types (thrombotic/hemorrhagic/Sub Arachnoid Hemorrhage/Cerebral Venous Thrombosis) will be examined as outcomes. All risk factors within the conceptual framework will be assessed for confounding during the Cox proportional hazard modeling. We propose to assess categorical representations in the event of non-linear relationships and develop binary definitions (e.g., presence/absence of hypertension, obesity, diabetes, stress, etc.) to facilitate model comprehensibility. The proportional hazard assumption will be tested using the score test for each risk factor. If the proportionality assumption is observed violated, Cox model for time varying covariates will be used for estimating the adjusted time varying hazard ratios (HRs). The HRs, their corresponding 95% CIs and p-values will be reported.

To develop gender specific stroke risk probability calculator, the estimated regression coefficients from the fitted gender specific Cox proportional hazards regression models will be used along with the estimated baseline survival functions at 5 years. All continuous predictors will be transformed using natural logarithms in order to minimize the influence of extreme outliers. The ability of the stroke risk probability in discriminating individuals who develop stroke from those who do not will be measured using c statistic [17]. The goodness of fit of the risk prediction model will be evaluated using a modified Hosmer and Lemeshow Chi square statistics which measures the model's ability of agreement between observed and predicted events within 5 years [18].

**Stroke and post-stroke period.** We intend to understand the effect of exposure groups from the Stroke-free period on survival, disability and quality of life among these stroke cases. Survival probabilities with corresponding 95% CIs at 28 days, 3, 6 and 12 months would be estimated using Kaplan Meier method. Univariate and multivariable Cox proportional hazard regression would be performed to identify the risk factors for death at 28 days, 3rd, 6th and 12th month. In the multivariable Cox proportional model, the confounders are identified based on the clinical importance as well as the difference in the baseline characteristics of the participants between survived and non-survived individuals at all four time points. The proportional hazard assumption will be tested using the score test for each risk factor. If the proportional assumption is observed violated, then Cox model for time varying covariates will be used for estimating the adjusted time varying hazard ratios (HRs). The HRs, corresponding 95% confidence intervals and p values will be reported.

The mRS is an ordinal scale with score ranging from zero to six. Proportional odds model will be used to identify the factors associated with disability at 28th day, 3rd month, 6th month and 12th month of onset of stroke. The quality-of-life variable is a continuous variable. Univariate and multiple linear mixed model regression analysis will be performed to identify the factors related with quality of life at 28th day, 3rd month, 6th month as well as after 12 months as outcome. In the mixed model, the subjects will be considered random and time will be considered as fixed. The linear or quadratic trend of quality of life over time will be checked using profile plots. Accordingly, time will be considered as a fixed continuous or categorical factor in the model. All confounders will be added as covariates in the model. The normality assumption for linear regression will be checked using Shapiro Wilk test as well as checking the histogram along with skewness and kurtosis. Appropriate transformations will be used if the dependent variable is found to be non- normal.

### Ethical clearance and consent

The study protocol was reviewed and approved by the Institutional Ethics Committee of NIMHANS vide letter NIMHANS/35[th] IEC (BS & NS DIV.)/2022; dated 17-06-2022. All eligible individuals shall be administered written informed consent. All concerned documents are available both in local language Kannada and English. The documents will be provided in the language preferred by the respondent/s. Illiterate individuals will have the consent form read to them in the presence of a literate witness and provide a thumb impression. All participants shall be given a copy of the signed informed consent form. The process of consent and data collection will be conducted at a place convenient to the participant ensuring adequate privacy and confidentiality. All study subjects detected to have any health issues during data collection, shall be informed about their health status and advised to seek appropriate care. All laboratory reports shall be sent to the respondent through an automated electronic reporting system to the phone numbers provided during data collection.

## Discussion

This large-scale community-based prospective-retrospective cohort study is envisaged to assess the effect of known and neo-risk factors in the natural history of stroke beginning from the stroke-free period to development of stroke and its subsequent outcomes (survival, disability and quality of life post-stroke). This study enrolls a sample of ~200,000 individuals naturally stratified into multiple population-based risk groups within Bengaluru and Kolara districts in Karnataka, India. Stratification of the population into six risk/exposure groups of single, dual, or multiple combinations for stroke will be done. At the end of the study, we propose to develop an India-specific stroke risk predictability calculator for the studied risk factors that shall predict an individual's risk of stroke over five-year periods. Further, data from this study shall also enable understanding the effect of these risk factors, individually and combined, on NCDs other than stroke and their outcomes.

Cohort study is the most suitable study design to accomplish our stated objectives. However, prolonged latency period of development of stroke in an individual's life span, multifactorial causation, and indefinite time of onset of risk factors/NCDs are known difficulties to conduct cohort studies on NCDs including stroke [19]. The employed mixed (Prospective-Retrospective) cohort design attempts to address these challenges. The study considers various apriori risk factors i.e., weight, mental health/stress, NCDs, nutrition, physical inactivity and substance use. Recall based retrospective /documentary evidence of the onset of certain risk factor counters the prolonged latency and provides valid information on the time of onset of risk factors. These study subjects are further followed prospectively to have a clear insight of the time period between onset of risk factors and development of stroke. Other major advantage of our mixed cohort study design is ability to study multiple exposures and multiple outcomes namely stroke, survival, disability, quality of life, and other NCDs. Cohort studies, mostly assess single exposures and multiple outcomes [19]. The mixed cohort study design in this scenario allows assessment of multiple exposures as well. Stroke being a rare event [20], a case control design is a better suited design [21]. However, concerns with respect to temporality, recall bias, and availability of funding made us opt for a mixed (Prospective-retrospective) cohort study. Other advantages like less susceptibility to selection bias, possibility to estimate true risk, measures of impact (population attributable risk and attributable risk) made us opt for cohort design compared to a case-control design. Furthermore, our objective to assess multiple exposures and outcomes along with developing a stroke-risk predictability calculator makes cohort study a valuable study design in comparison to other available study designs. Alternatively, stroke registries are an invaluable resource to assess the burden of stroke. Stroke

registries in India are predominantly urban-centric and limited to few risk factors and/or outcomes assessed [22, 23]. Difficulties in obtaining informed consent, unwillingness to share data, difficulty in retrieving medical records, and inadequate documentation of deaths are all known issues [24] limiting the utility of stroke registries to accomplish our stated objectives. Further, our study involves follow-up of naturally existing risk factor groups from the stroke-free period to the development of stroke and beyond, looking at stroke outcomes. This life course approach of assessing the effect of risk factors on stroke and its outcomes is unique to our study and is best accomplished by the adapted mixed cohort study design.

Ascertainment of risk factors and confounders, in our study, involves utilization of multiple validated and standardized tools for use within the community coupled with strong quality assessment and monitoring of study processes. Most tools proposed are extensively used routinely in clinical and biomedical research, medical practice, auditing and policy making [10, 25–42]. Age of onset/diagnosis of risk factors is considered to panelize data in this study, instead of the more typical and accurate time of assessment of the risk factor during the study [43]. The Cox model implicitly matches subjects on the time scale used. Since the rates of stroke increase with age, more so after onset of risk factor, it is assumed to probably have a larger effect than time of assessment. Thus, age of onset/diagnosis is considered most appropriate to use as time scale for the model.

One of the challenges in implementing such large-scale community-based cohort studies is ensuring community participation and high response rates [44]. Our study employs a strong community engagement plan to ensure high response rate and community participation. These measures are intended to reduce non-response/ loss-to-follow up–known limitations of a cohort study. The study team is acquainted with the cultural and social ethos of the study sites along with a long working relationship with the respective district administrations within Bengaluru and Kolara is likely to facilitate reduction in non-response.

Uniform and standardized data collection, minimizing known information biases like rounding errors, social desirability, and recall issues related to the exact dates of onset / diagnosis of risk factors assume importance when such large-scale cohort studies are implemented. Appropriate informed consent procedures, conducting interviews at a place convenient to the participant ensuring confidentiality and privacy are likely to reduce information bias. Gender-matched data collectors facilitate minimizing information bias mainly social desirability related to certain sensitive information.

Eligible respondents are those aged ≥30 years. This is based on the recommendation of the EAG since the Indian National Programme for Prevention & Control of Non-Communicable Diseases (NP-NCD) recommends opportunistic screening for risk factors and stroke among individuals aged 30 years and above [45]. Further, including these individuals shall capture dynamics of stroke in the young, an issue in countries of South-east Asian region, especially India [22, 23]. The study also includes pregnant and lactating women since they are known to develop stroke due to various risk factors, including those related to pregnancy [23].

The sample size for both stroke-free and post-stroke period is adequate in terms of the size and power. Although, *apriori* risk factors are known risk factors for stroke, to our knowledge, there are very minimal studies looking at these risk factors on stroke outcomes namely, survival, disability and quality of life. Not many studies assess the effect of various combinations of stroke risk factors on development of stroke and stroke outcomes. Our study provides an opportunity to understand risk factors of stroke, outcomes of stroke and various combinations of the same. This is a unique strength of our study.

The study employs specific quality assessment procedures along with an IMT, monitoring different processes involved in the study regularly. The entire study methodology is validated and approved by an EAG. In addition, the project team will monitor various activities and

processes, including data collection, data storage, training, and documentation focusing on quality. Strict protocols are established for data transfer and management with access-controlled mechanisms. All these efforts likely ensure implementation of study activities to the highest of standards.

## Strengths and limitations

This study has several strengths and certain limitations that need mention. The large sample size of ~200,000 individuals is a strength of the study. This serves to estimate the effect of individual and multiple combinations of risk factors to be assessed on stroke. Five among the six risk factors being assessed in this study are known risk factors. One of the risk factors (mental health/ stress cohort) is unique and a strength of the study. There are other neo-risk factors namely of gambling, COVID-19 infection and vaccination, sleep (Insomnia and obstructive sleep apnea), exposure to media, risk of cell phone addiction and restless leg syndrome being assessed in our study; which are unique & strength of the study. The study setting includes both urban (both slum and non-slum) and rural settings in Bengaluru and Kolara, which reflects the diversity and heterogeneity of the Indian population. Using a mixed cohort design (retrospective-prospective), allows for understanding the age of onset/diagnosis and for quantification of risk factors from onset/diagnosis. This enables assessment of time and dose response relationship between the risk factors, stroke and stroke outcomes. Thus, enabling comprehensive understanding of natural history of stroke across the lifespan, from the stroke-free period to outcomes of stroke. Development of stroke risk predictability calculator is a strength of this study. There are no such calculators available in India. Despite the strengths of this study, there are certain limitations that needs mention. Although data collectors would have been trained to collect information on certain sensitive issues, social desirability bias in reporting certain sensitive information like substance use cannot be completely ruled out. The adapted study design is known to increase the period of observation and negate the long latency associated with development of non-communicable outcomes. While ascertaining exposures, it is ideal to have time of onset as the beginning of exposure. However, date of diagnosis was the best alternative available given the elusive nature of onset of NCDs. This might pose a challenge of lead time bias associated with development of stroke as well as its consequent survival. This will be assessed and adjusted for in the analysis stage [46]. Adaptation of positive behaviors or change in risk behaviors are known challenges while conducting cohort studies [19]. This may reduce the incidence and risk of stroke among the study groups. However, sample size is calculated on the lowest possible incidence of stroke based on incidences reported in literature. This is likely to counter the reduced incidence and risk of stroke to a certain extent due to adoption of positive behavior. Intermixing of risk factors might complicate ascertaining measures of effect and impact. However, we propose to calculate rate ratio as the measure of effect. This negates the contamination of exposure ascertainment due to inter-mixing/changing of exposure status over the study period. The project is limited to specific areas, namely one taluka of Kolara district and two selected urban wards of Bengaluru city. Due to this limited coverage, the project may not fully statistically represent the diverse population and stroke burden across the entire state or country limiting the generalizability of results. But the choice of the study setting has its significance. The residents of the study sites come from various socio-demographic backgrounds (urban-rural-linguistic-slum/non-slum backgrounds). This ensures representation of various socio demographic/ economic and cultural strata representing diversity of the state or country. However, this might need to be further explored to ensure generalizability.

## Conclusion

This large population-based stroke cohort assessing stroke and its outcomes, first endeavor of its kind, would help address the need to have sufficient epidemiological data regarding the natural history of stroke, its management and outcomes for evidence-based stroke care in India. Being one of the largest population-based cohort studies in India the NIMHANS-NH-SKAN project has the potential to offer valuable information about stroke epidemiology, risk factors, and care in India with major implications in policy making and health programming both at community and national level including introduction of stroke risk predictability calculator in Indian National Programme for Prevention & Control of Non-Communicable Diseases (NP-NCD) in India in specific and similar countries in general.

## Supporting information

**S1 Table. Definitions proposed to ascertain exposures and outcomes for stroke-free period, stroke and post stroke period along with standardized tools.**
(DOCX)

## Acknowledgments

The authors extend their sincere gratitude to the Expert Advisory Group, whose invaluable insights and expertise significantly contributed to the development of this scientific paper. Special thanks to Dr. Pratima Murthy, Dr. Jeyaraj Durai Pandian, Dr. Sreekumaran Nair, Dr. Prashant Mathur, Dr. Srinivasa G A, Dr. Girish Baburao Kulkarni, Mr. Sundar Ramaswamy, and Dr. G Gururaj for their guidance and support. Additionally, the Independent Monitoring Team comprising of Dr. Akshaya KM, Dr. Usha S, Dr. Ramesh Holla, Dr. K Vidusha, Dr. Aravind Karinagannanavar, Dr. Malatesh Undi, Dr. Sharankumar Holyachi, and Dr. Anwith HS played a crucial role in ensuring the robustness and quality of the research. Their dedication and expertise are sincerely acknowledged. We would also like to thank the State and district health administration of Karnataka and Kolara respectively, health authorities of BBMP for providing administrative approvals for the conduct of the study. We would also like to thank our assistant research officers in our team Ms. Ashwini A and Shraddha S Pada.

## Author Contributions

**Conceptualization:** Pradeep S. Banandur, Gautham Melur Sukumar, Banavaram Anniappan Arvind, Srijithesh P. R., Binu V. S., Thimappa Hegde, Komal Prasad, Lavanya Garady.

**Data curation:** Pradeep S. Banandur, Gautham Melur Sukumar, Banavaram Anniappan Arvind, Binu V. S.

**Formal analysis:** Pradeep S. Banandur, Gautham Melur Sukumar, Banavaram Anniappan Arvind, Binu V. S.

**Funding acquisition:** Thimappa Hegde.

**Investigation:** Pradeep S. Banandur, Gautham Melur Sukumar, Banavaram Anniappan Arvind, Srijithesh P. R., Santosh Loganathan, Nandakumar Dalavaikodihalli Nanjaiah.

**Methodology:** Pradeep S. Banandur, Gautham Melur Sukumar, Banavaram Anniappan Arvind, Srijithesh P. R., Santosh Loganathan.

**Project administration:** Pradeep S. Banandur, Gautham Melur Sukumar, Banavaram Anniappan Arvind.

**Resources:** Pradeep S. Banandur.

**Supervision:** Pradeep S. Banandur, Gautham Melur Sukumar, Banavaram Anniappan Arvind.

**Validation:** Pradeep S. Banandur, Gautham Melur Sukumar.

**Visualization:** Pradeep S. Banandur, Srijithesh P. R., Binu V. S.

**Writing – original draft:** Pradeep S. Banandur, Gautham Melur Sukumar, Banavaram Anniappan Arvind, Srijithesh P. R., Binu V. S., Santosh Loganathan, Nandakumar Dalavaikodihalli Nanjaiah, Komal Prasad, Lavanya Garady, Akashanand Akashanand, Palak Poddar, Meenakshi Jayabalan, Isha Madan, Upashana Medhi, Arpitha Arun.

**Writing – review & editing:** Pradeep S. Banandur, Gautham Melur Sukumar, Banavaram Anniappan Arvind, Srijithesh P. R., Binu V. S., Santosh Loganathan, Nandakumar Dalavaikodihalli Nanjaiah, Thimappa Hegde, Komal Prasad, Lavanya Garady, Akashanand Akashanand, Palak Poddar, Meenakshi Jayabalan, Isha Madan, Upashana Medhi, Arpitha Arun.

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
