## [Decision Letter · Decision Letter 0]

18 Jul 2024

PONE-D-24-07473Population-based cohort across stroke life course in India-The NIMHANS-NH-SKAN Stroke project :  Study ProtocolPLOS ONE

Dear Dr. Banandur,

Thank you for submitting your manuscript to PLOS ONE. After careful consideration, we feel that it has merit but does not fully meet PLOS ONE’s publication criteria as it currently stands. Therefore, we invite you to submit a revised version of the manuscript that addresses the points raised during the review process.

**Thank you for your submission. Please respond carefully to reviewer 2 comments. Additional comments are also provided to improve the manuscript and scientific rigor of the proposed study. **

We look forward to receiving your revised manuscript.

Kind regards,

Dorothy Lall

Academic Editor

PLOS ONE

4. Please remove your figures from within your manuscript file, leaving only the individual TIFF/EPS image files, uploaded separately. These will be automatically included in the reviewers’ PDF.

Additional Editor Comments:

Thank you for this manuscript, the protocol is of an important study that would contribute to the understanding of stroke in India. However, we would like to suggest some changes for you to consider. These changes are suggested to improve the readability of the manuscript and to increase the rigor of the study in keeping with the reason we publish protocol papers.

Please carefully respond to the reviewer 2 comments, this will greatly improve the readability of the manuscript.

I want to raise some fundamental issues that may be good to incorporate or discuss in the manuscript

1. This is a cohort study- following up exposed but not diseased. In this, multiple exposure groups are described, including some novel risk/ exposed groups. An important consideration not discussed is lead time bias with regard to exposure. We cannot clearly know the onset of the exposure, detection of the risk factor is not the same as onset and this has bearing on interpretation of findings and hence should be discussed.

2. Please justify in some more detail why the cohort design is being used when the primary aim is to determine risk factors as opposed to the case control design. If however, the primary objective is to develop risk scoring and predictive calculators of incidence then this design is well justified. Please discuss this to justify the effort of a cohort study and the outcomes we can expect as readers.

In case developing predictive risk scores specific to our population is indeed an important objective, please describe in analysis how measurements upto 5 years will enable predictions for 10 year risk (the usual time frame for risk calculators). The 5 year duration seems to be less to fulfill the objective please provide some justification for this.

3. As risk prediction is an important aim of the study- please justify how urban poulation of Kolar and Bengaluru will enable applicability of the risk score across India.

4. Please do not name the health facilities for referral in the manuscript as it does not add to the understanding of the design; instead, please include a section on the setting and health care available that is more general and useful for readers not familiar with the Indian health care delivery system.

5. The description of the cohort study as retrospective and prospective is not clear- all cohort studies are prospective, all exposures have already occured. Kindly relook at description of the design. I also find the divison into stroke free and post stroke peiod rather artificial and inaccurate. There will be the population that were risk factor free to begin with but develop new risk factors or new combinations - it is rather dynamic something not addressed int the way the design is depicted. It is enough to mention that Follow up of cohort participants will be done till 12months after disease has occured.

6. Perhaps a section on the management of the cohort is useful- what efforts will be taken to keep the 2 lakh recruited participants within the cohort and minimise attrition. This increases accountability to the huge effort, time and money being invested in this research. The gains should justify the effort.

Reviewers' comments:

Reviewer's Responses to Questions

**Comments to the Author**

1. Does the manuscript provide a valid rationale for the proposed study, with clearly identified and justified research questions?

Reviewer #1: No

Reviewer #2: Yes

2. Is the protocol technically sound and planned in a manner that will lead to a meaningful outcome and allow testing the stated hypotheses?

Reviewer #1: No

Reviewer #2: Yes

3. Is the methodology feasible and described in sufficient detail to allow the work to be replicable?

Reviewer #1: No

Reviewer #2: Yes

4. Have the authors described where all data underlying the findings will be made available when the study is complete?

Reviewer #1: No

Reviewer #2: Yes

5. Is the manuscript presented in an intelligible fashion and written in standard English?

Reviewer #1: No

Reviewer #2: Yes

6. Review Comments to the Author

You may also provide optional suggestions and comments to authors that they might find helpful in planning their study.

Reviewer #1: This is an epidemiological study on stroke in Karnataka, involving NIMAHNS and other institutes. The aim is good and study worth perusing. However it fall outside my area of expertise hence I recommend that an epidemiologist should be consulted.

Reviewer #2: In this manuscript the authors describe a protocol for a population-based cohort study in the Karnataka state of India. I have following suggestions to improve the manuscript-

1. The abstract needs to be crisp. The background should list the importance of stroke as a public health problem, what are the current knowledge gaps that the study tries to fill and the key aims of the project.

2. It is unclear in the methods part of the abstract as to how the geographic areas (two districts) were selected for the study. Also, it is unclear if certain blocks from the two districts will be selected as the population of 200,000 doesn’t appear to be the population of the two districts.

3. Also, the authors should specify how the participants will be recruited in the cohort.

4. Please edit the sentence in the abstract- “is the first-of-its-kind, aims to fill critical gaps in epidemiological data concerning”

5. In the discussion part of the abstract the authors may want to use the words ‘important insights’ instead of ‘invaluable insight’. It is better to highlight the uniqueness of the study other than it is the largest cohort study and what specific insights this study will provide which have not been provided by many epidemiological and risk factors studies related to stroke in India.

6. The authors may want to remove these more generic statements in the abstract “This research has the potential to provide invaluable insights into stroke epidemiology, risk factors, and healthcare. The findings will carry significant implications for policy-making and health programming at both community and national levels.”

7. Introduction- “Stroke is one among the top five NCDs in the world (2)”, the authors may want to specify what do they mean by top five NCDs. For example they may specify that it is one of the top five leading causes of deaths worldwide.

8. In figure 1 specify how communities are selected i.e. how two wards in Bengalure and a block in Kolar district were selected.

9. It is unclear why “low risk group” is listed along with the risk factors and what does a bi-directional arrow between “physical activity” and “low risk group” suggests.

10. In figure 1, the second part may be better labelled as “stroke and post-stroke period” instead of just “post-stroke period”

11. Will medication-compliance be monitored in the post-stroke period?

12. In the methods part of the manuscript, please specify how the wards and sub-districts were selected.

13. Please provide a geographical map of study areas in two districts.

14. What processes will be in place to capture any missing stroke patients? The authors may want to specify. Do they have any idea about the potential percentage of patients who might be missed with their proposed method of enrolment?

15. In the “Ascertainment of exposure and outcome” section the authors mention that they will evaluate “tobacco dependence” Will they evaluate only tobacco dependence or tobacco use as well?

16. Kindly specify how current alcohol and tobacco use as mentioned in Table 3 will be defined.

17. In the same section the authors state that “certain neo-risk factors namely gambling, COVID-19 infection and vaccination, sleep (insomnia and obstructive sleep apnea), exposure to media, risk of cell phone addiction and Restless leg syndrome constitute exposures” . It will be good to provide operational definitions for each of these risk factors in an appendix.

18. Table 2 can be moved to an appendix.

19. Will blood be collected only for glucose and lipid estimations or other tests will also be done?

20. On page 33, the statement “Stroke will be examined for overall as well as type (thrombotic/hemorrhagic/SAH/CVT)” is unclear.

21. Table 3 has a back box. Why is it backed out?

22. Please check if all the references are formatted as per the journal’s requirements. I do not see them being appropriately formatted.

7. PLOS authors have the option to publish the peer review history of their article (what does this mean?). If published, this will include your full peer review and any attached files.

Reviewer #1: No

Reviewer #2: No

---

## [Author Response · Author response to Decision Letter 0]

14 Aug 2024

1.This is a cohort study- following up exposed but not diseased. In this, multiple exposure groups are described, including some novel risk/ exposed groups. An important consideration not discussed is lead time bias with regard to exposure. We cannot clearly know the onset of the exposure; detection of the risk factor is not the same as onset and this has bearing on interpretation of findings and hence should be discussed. 

Thank you for your keen observation. This is now addressed in the discussion section. 

2. Please justify in some more detail why the cohort design is being used when the primary aim is to determine risk factors as opposed to the case control design. If however, the primary objective is to develop risk scoring and predictive calculators of incidence then this design is well justified. Please discuss this to justify the effort of a cohort study and the outcomes we can expect as readers.

In case developing predictive risk scores specific to our population is indeed an important objective, please describe in analysis how measurements upto 5 years will enable predictions for 10-year risk (the usual time frame for risk calculators). The 5-year duration seems to be less to fulfill the objective please provide some justification for this. 

Thank you for this important observation. We have added a few points in discussion to provide more detail about the choice of cohort study design in comparison to case control design. 

With regards to developing risk predictability calculator, it may kindly be noted that the period of observation is not five years. The period of observation shall be the person-years since onset/diagnosis of a particular risk factor. This observation from the onset/diagnosis is in-built in the retrospective-prospective design adapted in the current study. Given this scenario, we are confident that it is possible to calculate both 5-year and 10-year risk for stroke.

As detailed in the discussion section, we acknowledge that a case-control design is often considered suitable for investigating rare events like stroke due to its efficiency in studying multiple risk factors with fewer subjects. However, after thorough deliberation and consultation, we opted for a mixed (prospective-retrospective) cohort study.

All the above has been addressed in the discussion section. This has now clearly enriched our manuscript. We sincerely thank the reviewers for the same.

3. As risk prediction is an important aim of the study- please justify how urban population of Kolar and Bengaluru will enable applicability of the risk score across India.

Thank you for this suggestion/comment. This particular observation is discussed in the limitations section. We have added a few more points to specifically address this suggestion. 

4. Please do not name the health facilities for referral in the manuscript as it does not add to the understanding of the design; instead, please include a section on the setting and health care available that is more general and useful for readers not familiar with the Indian health care delivery system. 

Thank you for this suggestion/comment. We have briefly explained about healthcare system in India by adding a section named “Setting and healthcare available”.

5. The description of the cohort study as retrospective and prospective is not clear- all cohort studies are prospective; all exposures have already occurred. Kindly relook at description of the design. I also find the division into stroke free and post stroke period rather artificial and inaccurate. There will be the population that were risk factor free to begin with but develop new risk factors or new combinations - it is rather dynamic something not addressed int the way the design is depicted. It is enough to mention that Follow up of cohort participants will be done till 12 months after disease has occurred. 

As indicated in the comment, we agree that all cohort studies are prospective because direction of enquiry is from exposure to the outcome.

Considering the long latency between exposures and occurrence of outcome, it is not viable to have exposure follow-up periods for 20-30 years. In this context, the cohort members are asked retrospectively of their exposures. But from the point of recruitment, we will follow for a period of 5 years where the enquiry of exposures is prospective. Henceforth, it’s a combination of retrospective and prospective (ambispective) exposure ascertainment. Thus, we feel that the description of cohort study as retrospective and prospective (ambispective) is appropriate.

We agree with the reviewers that exposures are dynamic. This is discussed in the discussion section along with limitations and strategy adapted to negate the same. We would also like to reiterate that the study follow-up during the stroke-free period is five years and post-stroke is 12 months. This is mentioned in the article.

Thank you for your insightful feedback on our manuscript. We appreciate your comments regarding the description of our cohort study design.

6. Perhaps a section on the management of the cohort is useful- what efforts will be taken to keep the 2 lakh recruited participants within the cohort and minimise attrition. This increases accountability to the huge effort, time and money being invested in this research. The gains should justify the effort. 

Thank you for this observation. We have now included the same.

7. The abstract needs to be crisp. The background should list the importance of stroke as a public health problem, what are the current knowledge gaps that the study tries to fill and the key aims of the project. 

The background part is now made crisp including the current knowledge gaps and key aims of the project. 

8. It is unclear in the methods part of the abstract as to how the geographic areas (two districts) were selected for the study. Also, it is unclear if certain blocks from the two districts will be selected as the population of 200,000 doesn’t appear to be the population of the two districts. 

We would like to clarify that the two districts were chosen based on operational convenience and the strong presence of our institute in these areas. Additionally, we selected two wards from the Bengaluru district and one sub-district from the Kolar, which together account for a total eligible population of 200,000. This is mentioned in the methods sections of the main manuscript. It is now mentioned in the abstract as well.

9. Also, the authors should specify how the participants will be recruited in the cohort. 

Thank you for your insightful comment. We appreciate your thorough review of our manuscript. Recruitment related information is included in the abstract.

10. Please edit the sentence in the abstract- “is the first-of-its-kind, aims to fill critical gaps in epidemiological data concerning” 

This is now edited.

11. In the discussion part of the abstract the authors may want to use the words ‘important insights’ instead of ‘invaluable insight’. It is better to highlight the uniqueness of the study other than it is the largest cohort study and what specific insights this study will provide which have not been provided by many epidemiological and risk factors studies related to stroke in India. 

Thank you. This is now addressed.

12. The authors may want to remove these more generic statements in the abstract “This research has the potential to provide invaluable insights into stroke epidemiology, risk factors, and healthcare. The findings will carry significant implications for policy-making and health programming at both community and national levels.” 

The lines are now removed based on reviewer’s suggestion. The discussion part of abstract is now edited. We thank the reviewers/editors for supporting us to enrich the abstract of the paper.

13. Introduction- “Stroke is one among the top five NCDs in the world (2)”, the authors may want to specify what do they mean by top five NCDs. For example, they may specify that it is one of the top five leading causes of deaths worldwide. 

Thank you for your valuable feedback. We have revised the sentence in the Introduction for clarity. The updated sentence now reads: "Stroke is the second leading cause of death and the leading cause of disability." We believe this change accurately reflects the significance of stroke as highlighted in global health statistics.

14. In figure 1 specify how communities are selected i.e. how two wards in Bengaluru and a block in Kolar district were selected. 

We are not sure about this comment and the relevance of including this information in figure 1. The information requested is included in the methodology section of the manuscript. 

15. It is unclear why “low risk group” is listed along with the risk factors and what does a bi-directional arrow between “physical activity” and “low risk group” suggests. The "low risk group" represents participants who do not belong to any of the other specified risk groups and serve as the non-exposed group (reference group).

We sincerely appreciate this keen observation of the reviewer. We have now removed the bidirectional arrow between “physical activity” and “low risk group” in figure 1. 

16. In figure 1, the second part may be better labelled as “stroke and post-stroke period” instead of just “post-stroke period” 

Thank you. We have renamed it as mentioned and appropriately edited in the remaining part of the manuscript.

17. Will medication-compliance be monitored in the post-stroke period?

Yes. Medication compliance will be monitored in the post-stroke period. This has now been articulated and incorporated in table 1 of manuscript.

18. In the methods part of the manuscript, please specify how the wards and sub-districts were selected. 

In the “methods” section of the manuscript, we have now specified that the wards and sub-districts were selected based on operational convenience. Specifically, the two wards in Bengaluru are adjacent to NIMHANS, which facilitates logistical ease for our research team. Additionally, these wards were chosen to capture diverse socio-economic settings: one ward includes a slum area, and the other comprises high and middle-class residents.

Our public health observatory is located in Kolar, making it a strategic choice for the study. We selected Mulbagal taluka as it represents a mix of rural (80%) and semi-urban (20%) populations, providing a comprehensive understanding of different community dynamics within the sub-districts of Kolar.

Our strong presence and established rapport within these communities will enable easy acceptance of the survey and effective community mobilization. This approach is expected to result in higher response rates and minimal lost to follow-up. This is articulated in the manuscript.

19. Please provide a geographical map of study areas in two districts.

Thank you for your valuable feedback. We are happy to provide a geographical map of the study areas in the two districts as per your suggestion. However, we would appreciate some clarification on the necessity of this addition. Additionally, including this map will cause us to exceed the maximum number of figures allowed by the journal. Please let us know how we should proceed in this case.

20. What processes will be in place to capture any missing stroke patients? The authors may want to specify. Do they have any idea about the potential percentage of patients who might be missed with their proposed method of enrolment? 

This is mentioned in the “identification and recruitment” section of the manuscript. We suspect minimal loss to follow-up since the stroke patients are from the community where the study is conducted. The community engagement established in place and the strong presence of the institute in the study area ensures maximum possible recruitment. In addition, those who have migrated from the study areas shall be attempted to contact through telephone and interviewed. This is mentioned in the “follow-up” section of the manuscript. Stroke incidence is known to be anywhere between 150-200 cases per lakh population. However, in the manuscript, the expected prevalence is conservatively taken as 150 considering missing of cases. 

21. In the “Ascertainment of exposure and outcome” section the authors mention that they will evaluate “tobacco dependence” Will they evaluate only tobacco dependence or tobacco use as well?

Yes. We intend to evaluate both use and dependence of tobacco. To avoid this confusion, we have now deleted the word ‘dependence’ from the section. 

22. Kindly specify how current alcohol and tobacco use as mentioned in Table 3 will be defined. 

This is defined in table 2 of the manuscript and hence was not mentioned in table 3 to avoid redundancy.

23. In the same section the authors state that “certain neo-risk factors namely gambling, COVID-19 infection and vaccination, sleep (insomnia and obstructive sleep apnea), exposure to media, risk of cell phone addiction and restless leg syndrome constitute exposures”. It will be good to provide operational definitions for each of these risk factors in an appendix. 

Thank you for your valuable feedback. We have provided the operational definitions of neo-risk factors in table 2 which is now moved to supporting information. Refer response to comment 24 below.

24. Table 2 can be moved to an appendix.

Thank you. We have moved it and appropriately referenced in the article.

25. Will blood be collected only for glucose and lipid estimations or other tests will also be done? 

Currently, blood samples will be collected specifically for glucose and lipid estimations as mentioned in “Biochemical sample collection and measurements” section. However, we plan to store the remaining blood for up to seven years to potentially run additional tests in the future if needed.

26. On page 33, the statement “Stroke will be examined for overall as well as type (thrombotic/hemorrhagic/SAH/CVT)” is unclear. 

Thank you for your valuable feedback. The updated sentence now reads “Stroke as well as its sub types (thrombotic/hemorrhagic/Sub Arachnoid Hemorrhage/Cerebral Venous Thrombosis) will be examined as outcomes.”

27. Table 3 has a black box. Why is it blacked out?

Thank you for this observation. The table is now reformatted without the black box for clarity.

28. Please check if all the references are formatted as per the journal’s requirements. I do not see them being appropriately formatted.

It is now formatted as per the journal’s requirements.

---

## [Editor Report · Decision Letter 1]

29 Aug 2024

Population-based cohort across stroke life course in India-The NIMHANS-NH-SKAN Stroke project : A Study Protocol

PONE-D-24-07473R1

Dear Dr. Banandur,

We’re pleased to inform you that your manuscript has been judged scientifically suitable for publication and will be formally accepted for publication once it meets all outstanding technical requirements.

Kind regards,

Dorothy Lall

Academic Editor

PLOS ONE

Additional Editor Comments (optional):

Thank you for thoughtfully responding to the comments. About including maps, please consider including as supplementary material, though not mandatory to include.
---

## [Editor Report · Acceptance letter]

20 Sep 2024

PONE-D-24-07473R1 

PLOS ONE

Dear Dr. Banandur, 

I'm pleased to inform you that your manuscript has been deemed suitable for publication in PLOS ONE. Congratulations! Your manuscript is now being handed over to our production team.

Kind regards, 

on behalf of

Dr. Dorothy Lall 

Academic Editor

PLOS ONE